# Aerobic glycolysis supports hepatitis B virus protein synthesis through interaction between viral surface antigen and pyruvate kinase isoform M2

Yi-Hsuan Wu[1], Yi Yang[1], Ching-Hung Chen[1], Chia-Jen Hsiao[1,2], Tian-Neng Li[1], Kuan-Ju Liao[3], Koichi Watashi[4], Bor-Sen Chen[5], Lily Hui-Ching Wang[1] *

1 Institute of Molecular and Cellular Biology, National Tsing Hua University, Hsinchu, Taiwan, 2 Division of Gastroenterology, New Taipei City Hospital, New Taipei City, Taiwan, 3 Institute of Bioinformatics and Structural Biology, National Tsing Hua University, Hsinchu, Taiwan, 4 Department of Virology II, National Institute of Infectious Diseases, Tokyo, Japan, 5 Department of Electrical Engineering, National Tsing Hua University, Hsinchu, Taiwan

* lilywang@life.nthu.edu.tw

**Data Availability Statement:** All relevant data are within the manuscript and its Supporting Information files.

## Abstract

As an intracellular pathogen, the reproduction of the hepatitis B virus (HBV) depends on the occupancy of host metabolism machinery. Here we test a hypothesis if HBV may govern intracellular biosynthesis to achieve a productive reproduction. To test this hypothesis, we set up an affinity purification screen for host factors that interact with large viral surface antigens (LHBS). This identified pyruvate kinase isoform M2 (PKM2), a key regulator of glucose metabolism, as a binding partner of viral surface antigens. We showed that the expression of viral LHBS affected oligomerization of PKM2 in hepatocytes, thereby increasing glucose consumption and lactate production, a phenomenon known as aerobic glycolysis. Reduction of PKM2 activity was also validated in several different models, including HBV-infected HepG2-NTCP-C4 cells, adenovirus mediated HBV gene transduction and transfection with a plasmid containing complete HBV genome on HuH-7 cells. We found the recovery of PKM2 activity in hepatocytes by chemical activators, TEPP-46 or DASA-58, reduced expressions of viral surface and core antigens. In addition, reduction of glycolysis by culturing in low-glucose condition or treatment with 2-deoxyglucose also decreased expressions of viral surface antigen, without affecting general host proteins. Finally, TEPP-46 largely suppressed proliferation of LHBS-positive cells on 3-dimensional agarose plates, but showed no effect on the traditional 2-dimensional cell culture. Taken together, these results indicate that HBV-induced metabolic switch may support its own translation in hepatocytes. In addition, aerobic glycolysis is likely essential for LHBS-mediated oncogenesis. Accordingly, restriction of glucose metabolism may be considered as a novel strategy to restrain viral protein synthesis and subsequent oncogenesis during chronic HBV infection.

**Funding:** This study was supported by the Taiwan Ministry of Science and Technology (MOST-105-2320-B-007-006-MY3 and MOST-107-2221-E-007-107-MY3 to LW), National Tsing Hua University (NTHU-109Q2808E1, NTHU-109Q2713E1 to LW), and the Liver Disease Prevention & Treatment Research Foundation (Taiwan). The funders had no role in study design, data collection and analysis, decision to publish, or preparation of the manuscript.

**Competing interests:** The authors have declared that no competing interests exist.

## Author summary

Chronic HBV infection is a life-long threat of patients, with a 25~40% increased risk of developing liver cirrhosis and cancer. Persistent expression of oncogenic viral products in the liver, especially LHBS, is an oncogenic caveat and resistant to current antiviral agents. Here we show that viral surface antigens bind to host PKM2 and diminish its kinase activity. This virus-host interaction induces metabolic switch from oxidative phosphorylation to aerobic glycolysis, with increased glucose consumption and lactate production. We show that such metabolic switch not only favors viral protein synthesis but also contributes hepatocarcinogenesis. Notably, restoration of PKM2 activity by chemical activators decreases expressions of viral products and largely suppresses virus-mediated hepatocarcinogenesis. This study highlights the importance of host metabolism in supporting viral protein synthesis and indicates a novel therapeutic approach to control chronic HBV infection via modulating host metabolic switch.

## Introduction

Hepatitis B is a life-threatening infectious liver disease caused by the hepatitis B virus (HBV). Chronic HBV infection affects ~292 million people worldwide (63.8% in Asia and 27.6% in Africa) in 2016, and has potential adverse outcomes that include hepatic decompensation, cirrhosis and/or hepatocellular carcinoma (HCC) [1]. WHO reported that 27 million people (10.5% of all people estimated to be living with chronic hepatitis B) were aware of their infection, while 4.5 million (16.7%) of the people diagnosed were on treatment (WHO. Global Hepatitis Report 2017. Geneva: 2017 ISBN: 978-92-4-156545-5). Though the currently available antiviral drugs can effectively reduce serum viral load in patients with chronic hepatitis B, complete elimination of the virus in the liver is still difficult. The blame goes to the long-lasting nature of an intracellular viral replication intermediate termed covalently closed circular (ccc) DNA, which serves as a viral persistence reservoir in the liver [2].

The HBV virion contains a compact 3.2-kb genome that exists as a partially double-stranded, relaxed circular DNA (rcDNA). Upon infection, cccDNA is generated as a plasmid-like episome in the host cell nucleus from the viral rcDNA genome [3]. Viral cccDNA is the template for all viral transcripts, and in consequence of new virions. The virion comprises an outer envelope of the lipid-embedded small (S), middle (M) and large (L) surface antigens (HBS) and an inner nucleocapsid (core particle; hepatitis B core antigen/HBcAg in serology). The three surface antigens collectively comprise HBsAg in serology. The virus also produces precore protein, serologically known as e antigen, or HBeAg. Both HBsAg and HBeAg play essential roles in chronic infection and are used as serological biomarkers of clinical examinations. High HBeAg is thought to induce T cell tolerance to HBeAg and HBcAg, which may contribute to early viral persistence upon initial infection [4]. Serum HBsAg has two resources, infectious virions enveloped with HBsAg and large amounts of subviral particles consisting of HBsAg. High HBsAg is believed to contribute to T cell exhaustion, resulting in limited or weak T cells response and even deletion of HBV-specific CD4 and CD8 T cells during T cell differentiation [5].

For patients with chronic hepatitis B, the treatment endpoint is the loss of HBsAg, known as "functional cure", a state to indicate an effective control of HBV and long-term prognosis [6]. Spontaneous loss of HBsAg was detected in about 1.2% per year in treatment-naive patients in a recent systematic review and pooled meta-analyses reported [7]. It is noted that current antiviral therapies, such as type 1 interferons and nucleos(t)ide analogues (NUCs),

rarely provoked the loss of HBsAg (0~10%), even after prolonged treatment [8]. As cccDNA is the intracellular viral reservoir, current pipelines of new HBV therapeutics have been focused on the eradication of the cccDNA [9]. On the other hand, several lines of evidence demonstrated that HBsAg was expressed not only from the cccDNA but also from the viral DNA integrated into the host genome. A recent RNAi-based treatment of chronically infected patients and chimpanzees also revealed that integrated hepatitis B virus DNA is a source of viral HBsAg, and even the dominant source in HBeAg-negative chimpanzees [10]. As HBV DNA integration may occur immediately upon infection [11], the abundance of hepatocytes with integrated HBV DNA in the liver may set a high threshold for the functional cure. Notably, integrated viral DNA with naturally derived PreS-truncation has been linked to the development of liver cancer. Several lines of evidence have shown that LHBS carrying PreS mutants, especially the PreS2 mutant LHBS, are driving factors of genomic instability and subsequent tumorigenesis [12–14]. A recent study reported that patients with PreS2 mutant were at high risk of hepatoma recurrence after curative hepatic resection [13]. Accordingly, persistence expression of intrahepatic LHBS is an oncogenic caveat in patients with chronic hepatitis B.

Because the loss of HBsAg is the indicator of endpoint treatment, there is interest in directly reducing expression of viral antigens and regulatory proteins. This study therefore aims to test if host metabolism may be tailored to control viral biosynthesis in the case of HBV. To this end, we set up to find intrahepatic metabolic regulators that may interact with HBsAg. Interestingly, protein pyruvate kinase isoform M2 (PKM2) is identified from the affinity purification with viral LHBS. PKM2 is a master regulator in glycolysis and has been implicated as a major metabolic switch in cancer metabolism [15]. Our evidence indicated that LHBS reduced PKM2 activity and thereby increased overall glucose consumption and lactate production in hepatocytes. Interestingly, modulation of glucose metabolism, either via chemical activation of PKM2 activity or the reduction of glucose supply, reduced HBV biosynthesis without affecting general host proteins. These data therefore indicate that intrahepatic viral biosynthesis may rely on the metabolic switch of host metabolism.

## Results

### PKM2 is a binding partner of LHBS

To explore the interaction between LHBS and host proteins, we set up an affinity purification experiment to identify potential intracellular binding partners of LHBS. Previously established stable cell lines carrying SNAP-tagged wild type LHBS [12,16] were used for affinity purification and the resulting lysates were subjected to Mass-Spectrometry. PKM2 was identified in the SNAP-LHBS pulled-down lysates (S1 Table). To confirm the interaction between PKM2 and LHBS, we performed a reciprocal immunoprecipitation. As shown in Fig 1A, PKM2 was detected in the pulled-down lysate of LHBS. In addition, LHBS was detected in the pulled-down lysate of PKM2. To understand which subcellular localization that LHBS and PKM2 interaction was taking place, we applied the proximity ligation assay (PLA) technology [17]. This technology provoked signal amplification of two complementary oligonucleotides that were linked to distinct antibodies specific for LHBS and PKM2. Using PLA technology, we showed that a subset of PKM2 interacted with LHBS mainly in the cytoplasm, and displayed as dot-like signals in LHBS-expressing cells (Fig 1B).

To further confirm the interaction between PKM2 and native viral surface antigens, we transfected HuH-7 cells with a plasmid carrying a complete HBV genome (pHBV3.6) and performed immunoprecipitation using monoclonal and polyclonal antibodies specific for HBS. As all three surface antigens (L, M, and SHBS) contain the same C-terminus HBS region, the

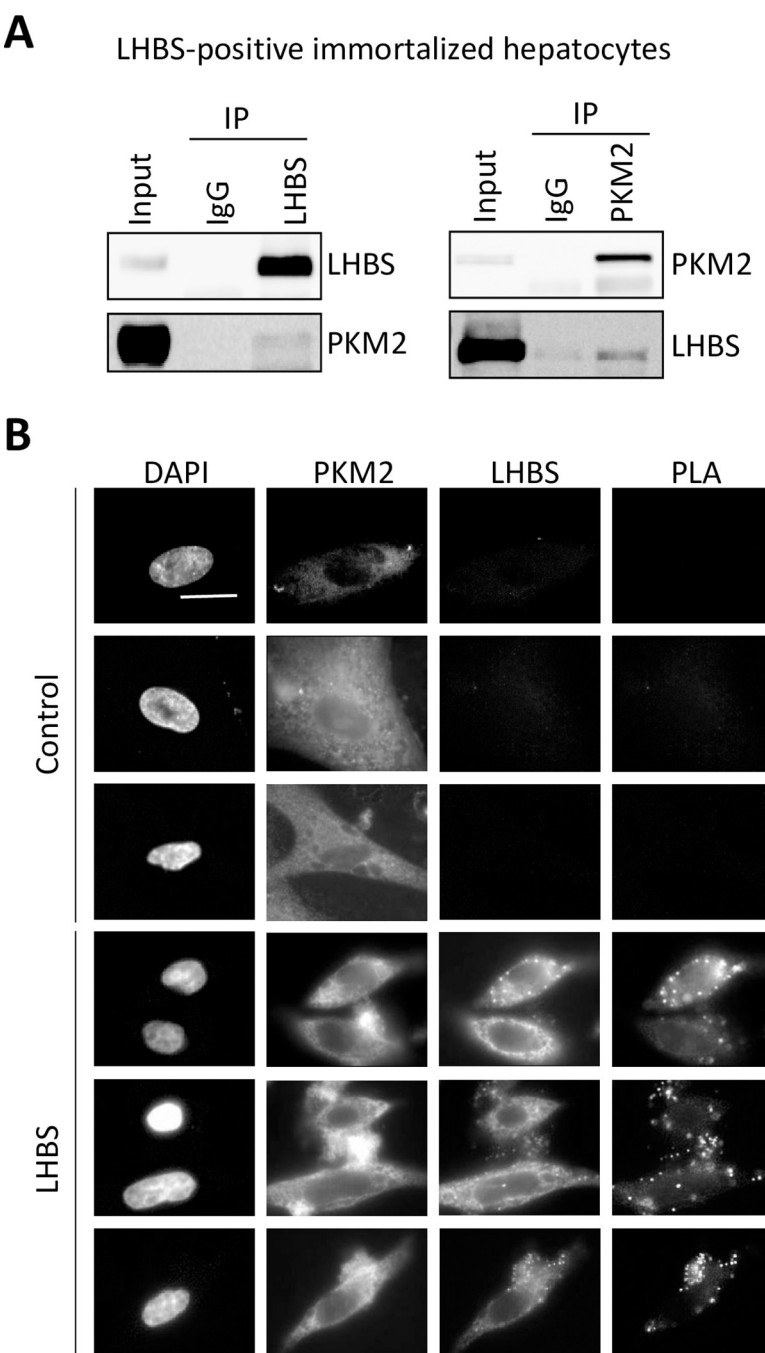

**Fig 1. PKM2 is a binding partner of viral LHBS.** (A) Reciprocal immunoprecipitations using control IgG, anti-LHBS (anti-PreS1, clone 7H11), and anti-PKM2 antibodies were carried out using total cell lysates prepared from an immortalized hepatocyte line stably expressing LHBS. Representative immunoblots from three independent experiments are shown. (B) Visualization of PKM2-LHBS interactions in immortalized hepatocytes by proximity ligation assay (PLA). Control and LHBS-positive cells were stained with primary antibodies against LHBS/PreS1 and PKM2, followed by incubation with Duolink PLA probes detecting LHBS/PKM2 interactions and fluorescent labeled secondary antibodies. The cell nuclei were stained with DAPI. Three representative images were shown for each category. Scale bar denotes 10 μm.

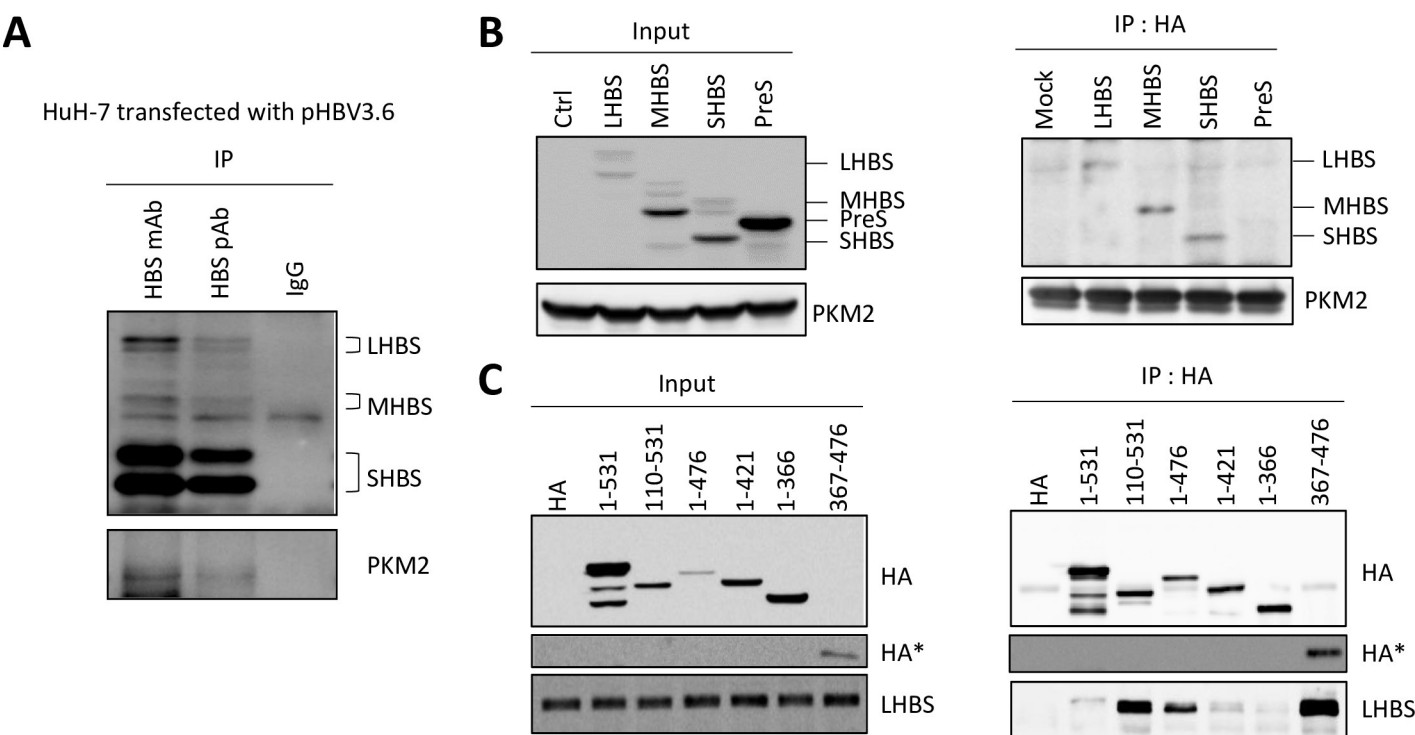

**Fig 2. Identifying the minimal binding region of LHBS and PKM2.** (A) Total cell lysates of HuH-7 cells transfected with pHBV3.6 plasmid were harvested for affinity immunoprecipitations using control immunoglobulin (IgG), monoclonal (mAb, clone 86H6) and polyclonal (pAb, Abnova) anti-HBS antibodies. The resulted immunoprecipitants were probed with anti-HBS mAb (86H6) and anti-PKM2 antibodies. Representative immunoblots were shown. (B) 293T cells were transiently transfected with expression constructs of HA-PKM2 and SNAP-tagged-LHBS, -MHBS, -SHBS, and -PreS, as indicated, for 2 days, and total cell lysates were harvested for affinity immunoprecipitation using a mouse anti-HA antibody. Representative immunoblots using anti-PKM2 and anti-SNAP antibodies were shown. (C) Immortalized hepatocytes stably expressing LHBS were transfected with expression constructs of HA-PKM2 encoding different truncation fragments as indicated. Total cell lysates were harvested at 2 days post transfection and subjected to affinity immunoprecipitation using anti-HA antibody. Representative immunoblots of HA-PKM2 and LHBS detected by anti-HA and anti-LHBS/PreS1 antibodies were shown. *denotes the high exposure image of the low molecular weight protein HA-367-476.

immunoprecipitation using anti-HBS antibodies successfully pulled-down all three surface antigens. PKM2 was co-precipitated with viral surface antigens by both monoclonal and polyclonal anti-HBS antibodies, but not with the control immunoglobulin (Fig 2A). To identify essential interacting domains of PKM2 and LHBS, we made different expression constructs including SNAP-tagged LHBS (PreS1+S2+S), MHBS (PreS2+S), SHBS (S only), and PreS (PreS1+S2). Upon co-transfection with HA-tagged PKM2 into 293T cells, we detected LHBS, MHBS, and SHBS in the pull-down lysate of HA-PKM2, but not PreS, indicating that HBS was the major binding domain of PKM2 (Fig 2B). We next generated several constructs of PKM2 and performed transient transfection on a stable cell line expressing SNAP-LHBS. Notably, whereas the full-length PKM2 weakly interacted with LHBS, HA-PKM2 110–531 showed a strong affinity with LHBS. This interaction was reduced upon serial truncation on the C-terminus, suggesting that C-terminus of PKM2 was important for binding to HBS. Finally, HA-PKM2 367–476 displayed a strong binding affinity with LHBS (Fig 2C). In short, we mapped minimal interaction domains to the HBS region of LHBS and the C-terminus amino acid 367–476 of PKM2. Taken together, as all surface antigens share the same C-terminus HBS, our results indicated that PKM2 interacts with all viral envelope proteins.

## LHBS provokes aerobic glycolysis via diminishing PKM2 activity

To study the interplay between PKM2 and viral surface antigens under native conditions, we applied an infectious model using HepG2-NTCP-C4 cells [18,19]. The infectious HBV virions were collected from the culture supernatant of HepAD38 and purified by sucrose cushion ultracentrifugation [20]. To confirm a successful HBV infection, we measured secretion of HBsAg and HBeAg following infection with HBV at 1000 multiplicity of infection (MOI). We pretreated HepG2-NTCP-C4 with 2.5% DMSO before infection and infected these hepatocytes with HBV virions in the presence of 4% PEG8000 and 2.5% DMSO. We collected culture supernatants and measured the levels of HBsAg and HBeAg secretion. A gradual increase in the extracellular secretion of HBsAg or HBeAg was shown in Fig 3A. Although the intracellular expression of SHBS was weak, we clearly detected the expression of viral LHBS upon HBV infection (Fig 3B).

As PKM2 is an important enzyme that catalyzes the last and physiologically irreversible step in glycolysis, we tested if PKM2 activity was affected by HBV. Upon infection of HBV on HepG2-NTCP-C4 cells, PKM2 activity was reduced by 30% (Fig 3C). In addition, PKM2 activity was reduced by 86% in LHBS-expressing immortalized hepatocytes (Fig 3D). As the reduction of PKM2 activity can be a result of reducing protein expression, we exclude this possibility by showing that the expression of PKM2 was not changed in LHBS-expressing hepatocytes (Fig 3E), as well as in HepG2-NTCP-C4 infected with HBV (Fig 3B).

In the cell, PKM2 may exist as either a low-activity dimeric or high-activity tetrameric state [21]. We then investigated whether PKM2 oligomerization is affected by LHBS on immortalized hepatocytes. To this end, cell lysates were treated without or with 0.1% glutaraldehyde (GA) for crosslinking and then subjected to western blotting. Noted that overall PKM2 expression levels were equal in both control and LHBS cells, as shown by the same level of monomer without GA crosslinking (Fig 3F, left). With GA crosslinking, endogenous levels of PKM2 dimer and tetramer were detected at two smear pattern signals at higher molecular weight on the blot. In the panel with crosslinking, PKM2 dimerization was increased by approximately 3 folds in LHBS cells, in comparison to the control line (Fig 3F, right). To further characterize the overall PKM2 oligomerization states in the cell, we performed glycerol density gradient centrifugation on total cell lysates of control and LHBS cells. As shown in Fig 3G, major population of PKM2 was detected in the high molecular weight tetrameric fractions spanning from 22 to 25 in control cells. An increase in low molecular weight PKM2 dimer fraction, spanning from 14 to 18, was detected in LHBS-expressing cells. Together, the expression of viral LHBS favored dimerization of PKM2 and thereby decreased the kinase activity of PKM2.

PKM2 is a key enzyme in the glucose metabolism. A reduction of PKM2 activity was shown to induce metabolic switch to aerobic glycolysis, a phenomenon known as Warburg effect, which provides cancer cells with growth advantages [15]. We next explored if the reduction of PKM2 activity may induce metabolic switch on LHBS-expressing hepatocytes. Here we showed that both glucose consumption and lactate production were significantly increased on immortalized hepatocytes expressing viral LHBS (Fig 3H).

As an additional support, we applied recombinant adenovirus carrying wild type HBV (Ad-HBV-WT) or control knockout (Ad-HBV-KO) genome as an *in vitro* model [22] and examined changes of PKM2 oligomerization by GA crosslinking on either HuH-7 or HepG2 cells. We first infected HuH-7 cells with Ad-HBV-WT or Ad-HBV-KO or a 1:1 mixture in 200 MOI for 2 days and collected total cell lysates for crosslinking analysis. As expected, LHBS expression increased with the dose of Ad-HBV-WT (Fig 4A). In addition, PKM2 dimerization increased following infection with Ad-HBV-WT in a dose-dependent manner, after normalization with total PKM2 as defined by the monomer signal in the non-crosslinking panels (Fig

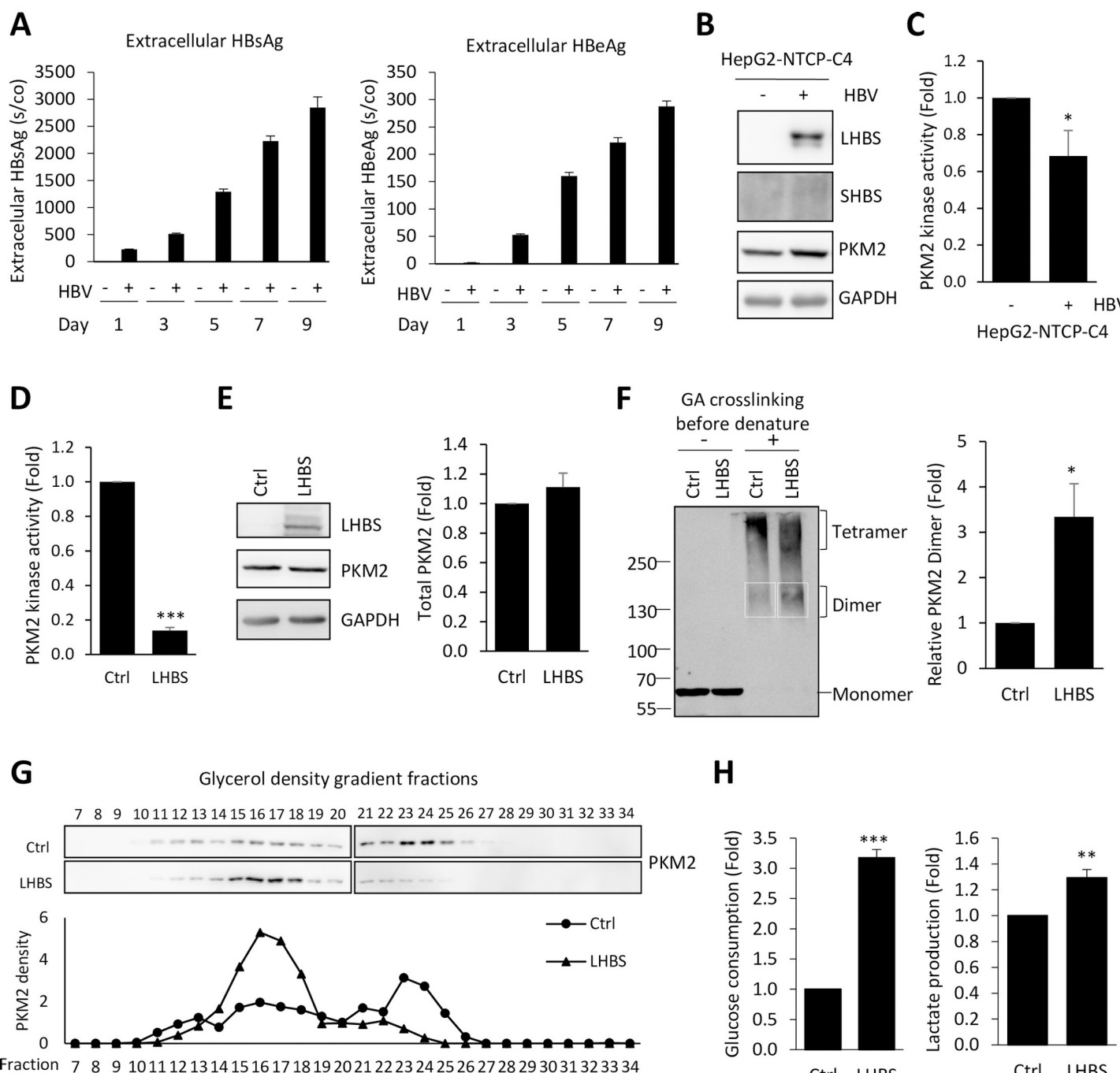

**Fig 3. HBV reduces PKM2 kinase activity and provokes aerobic glycolysis.** (A) HepG2-NTCP-C4 cells were infected with HBV virions at 1000 MOI in the presence of 4% PEG8000 and 2.5% DMSO. Secretions of viral HBsAg and HBeAg were measured in culture supernatants harvested at indicated time. Signal-to-cutoff (s/co) values of HBsAg and HBeAg were shown. (B) HepG2-NTCP-C4 cells were infected with HBV at 1000 MOI for 9 days and total cell lysates were harvested for the detection of LHBS, SHBS and PKM2 by western blotting. GAPDH expression was detected as an internal control. (C) HepG2-NTCP-C4 cells were infected with or without HBV at 1000 MOI for 9 days. Total cell lysates were collected for measuring PKM2 kinase activity. Relative PKM2 activities were shown after normalization with that of non-infected cells. (D) PKM2 activities measured in immortalized hepatocytes with or without LHBS expression. Relative PKM2 activities were shown after normalization with that of control cells. (E) Representative western blots showing expressions of LHBS and PKM2 on immortalized hepatocytes. The right panel showed relative quantitation of PKM2 after normalization with PKM2 intensity of control cells. GAPDH was detected as a loading control. (F) Total cell lysates of immortalized hepatocytes were treated with or without 0.1% glutaraldehyde (GA) for 20 min crosslinking before denatured for SDS gel electrophoresis and western blotting. Representative blot of PKM2 is shown. The right panel indicates relative signals of PKM2 dimer detected in the white box area on the blot, after normalization with total PKM2. (G) Glycerol density gradient centrifugations of PKM2 in control and LHBS-expressing immortalized hepatocytes. Total cell lysates were loaded on top of the 15–35% glycerol gradient and the samples were centrifuged at 50000 rpm for 16 hours at 4°C. Fractions were collected and subjected to western blotting with anti-PKM2

antibody. PKM2 density was calculated and plotted the graph against each fraction in the lower panel. (H) Glucose consumption and lactate production in the culture media of control and LHBS-expressing immortalized hepatocytes were measured. All quantitative results were a summary of three repeats and data were displayed as mean ± standard error. $^*p<0.05$, $^{**}p<0.01$, and $^{***}p<0.001$ were calculated using Student's t test.

4B). Similar results were observed in HepG2 cells infected with Ad-HBV-WT, except that 100 MOI was used for HepG2 (Fig 4C and 4D). Finally, we confirmed that kinase activity of PKM2 was reduced by 13% in HuH-7 cells infected with Ad-HBV-WT (Fig 4E). To further clarify whether expression of SHBS alone has the same effect as LHBS on modulating PKM2, we generated two HBV mutants of which the start codon of LHBS on the PreS1 region and start codon of MHBS on the PreS2 region were mutated by site-directed mutagenesis of pHBV3.6,

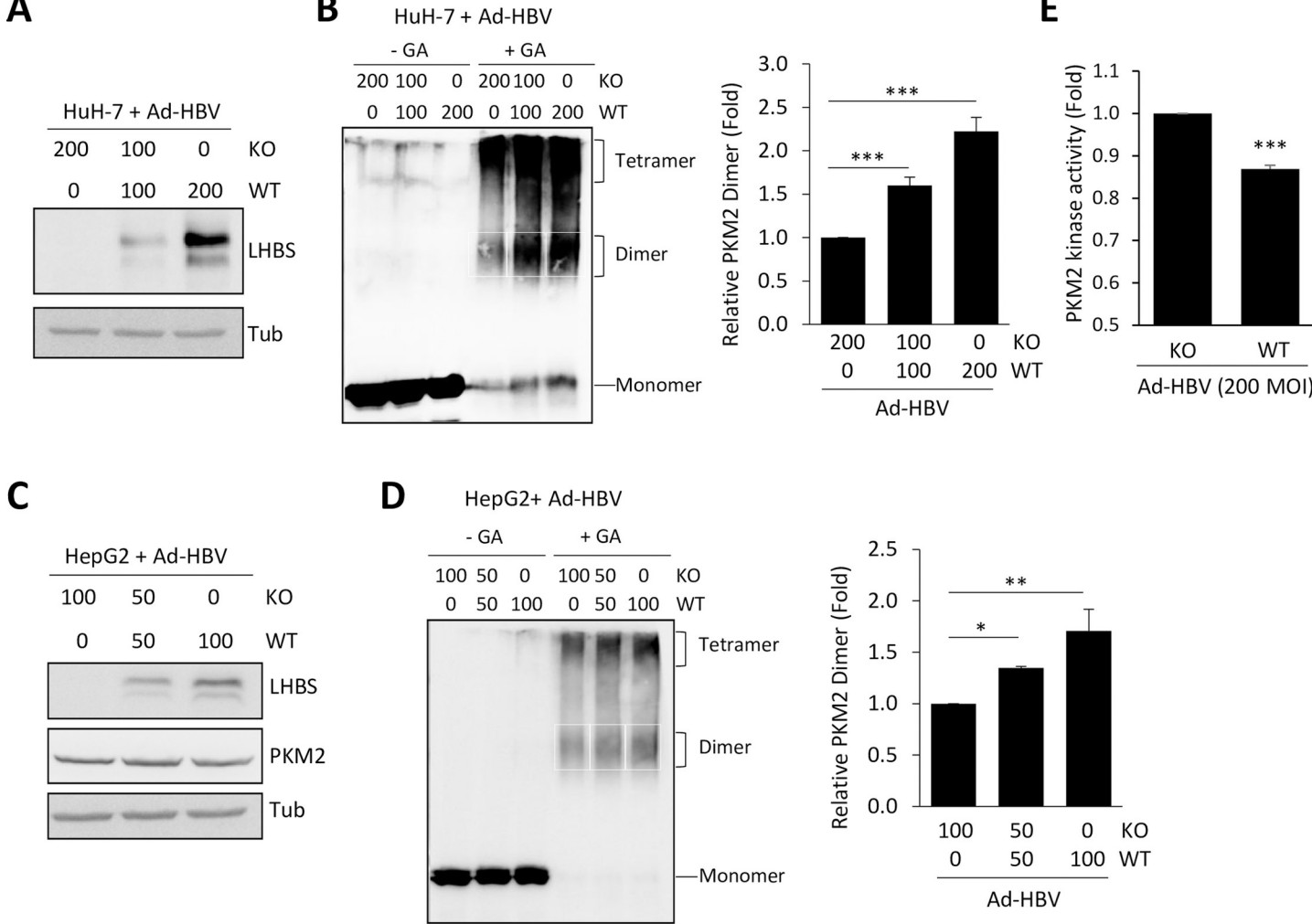

**Fig 4. Infection of Ad-HBV-WT increases PKM2 dimerization.** (A) HuH-7 cells were infected with either Ad-HBV-KO or Ad-HBV-WT at indicated MOI for 2 days. Representative blots of viral LHBS and tubulin were shown. (B) Total cell lysates of HuH-7 cells infected with Ad-HBV were collected and treated with or without glutaraldehyde (GA) crosslinking for 20 minutes. Representative blot of PKM2 is shown. The left panel showed relative PKM2 dimer in each treatment group. The quantity of dimeric PKM2 was defined by signal detected in white rectangular region of each sample, after normalization with total PKM2 detected in the non-crosslinking samples. (C) HepG2 cells were infected with either Ad-HBV-KO or Ad-HBV-WT at indicated MOI for 2 days. Representative blots of viral LHBS, PKM2, and tubulin were shown. (D) PKM2 dimerization in Ad-HBV infected HepG2 cells was measured by western blot shown on the left panel and quantitation results shown on the right panel. (E) HuH-7 cells were infected with Ad-HBV-KO or Ad-HBV-WT for 2 days and lysates were collected for measuring PKM2 activity. Relative PKM2 activities in Ad-HBV-WT infected cells were shown after normalization with that of Ad-HBV-KO infected cells. All quantitative results were displayed as mean ± standard error from three replicates. $^*p<0.05$, $^{**}p<0.01$, and $^{***}p<0.001$ were calculated using Student's t test.

i.e., a L-deficient mutant and a L/M-deficient mutant (S1 Text). Accordingly, these two HBV mutant constructs only allowed expressions of MHBS+SHBS, or SHBS alone on transfected hepatocytes. We found that PKM2 kinase activity was reduced in hepatocytes expressing MHBS+SHBS or only SHBS (S1 Fig). Thus, expression of SHBS alone was capable of reducing PKM2 activity in hepatocytes. Taken together, HBV envelope proteins facilitate PKM2 dimerization, diminish PKM2 kinase activity, and thereby provoke aerobic glycolysis in hepatocytes.

## PKM2 negatively regulates protein synthesis of HBV

As an intracellular pathogen, viruses may hijack host metabolism to support their own propagation. Specifically, viral infection may trigger metabolic reprogramming in host cells to facilitate optimal virus production [23]. We therefore suggested that the metabolic switch induced by LHBS might be essential for HBV biosynthesis. To test this hypothesis, we asked if activation of PKM2 activity may affect expression of HBV products in the host. Specifically, we transiently transfected pHBV3.6 plasmid into HuH-7 cells and treated with TEPP-46, a compound known to stabilize tetrameric PKM2 [24]. Upon treatment with TEPP-46, extracellular secretions of HBsAg and HBeAg were reduced by 40% and 20%, respectively (Fig 5A). In addition, intracellular expressions of LHBS, SHBS, and HBcAg were reduced by 30, 41, and 26%, respectively (Fig 5B and 5C). In contrast, expression levels of viral HBx and general host proteins, including GRP78, HSC70, PKM2, Plk1, Mad2L1, Bcl2, and PCNA, were not affected by TEPP-46.

Similar results were obtained in cells treated with DASA-58, another PKM2 activator [24]. PKM2 activity in LHBS cells was increased by DASA-58 (Fig 6A). As expected, extracellular secretions of HBsAg and HBeAg were also reduced (Fig 6B). We confirmed that intracellular expressions of viral LHBS, SHBS, and HBcAg were reduced upon DASA-58 treatment, except for viral HBx (Fig 6C). Next we asked if restoration of PKM2 activity may reduce intermediate metabolites of pentose phosphate pathway. This was confirmed by the reduction of 6-phosphogluconic acid (6-PGA) in cells treated with TEPP-46 and DASA-58 (Fig 6D). To examine the impact of PKM2 on the stability of viral products, we monitored protein stabilities in pHBV3.6 transfected HuH-7 cells. As shown in the S2 Fig, general stability of viral LHBS, SHBS, HBcAg, and HBx were not significantly changed upon treatments of TEPP-46 or DASA-58. In addition, HBV RNA was not affected upon PKM2 activation as shown in the S3 Fig. Finally, we measured production of HBV virions in HBV-infected HepG2-NTCP-C4 cells in the presence or absence of chemical PKM2 activators. Production of HBV virion was measured by detecting viral load in the culture supernatant using quantitative PCR. We found that HBV viral load was reduced by approximately 20% upon treatment of TEPP-46 and DASA-58. Noted that only the reduction of HBV viral load upon TEPP-46 treatment was significant after statistical analysis (Fig 6E). Taken together, we conclude that synthesis of major HBV products was suppressed upon restoration of PKM2 activity.

## Productive HBV protein synthesis relies on aerobic glycolysis

To explore if viral-specific protein synthesis can be regulated by modulating glucose metabolism, we cocultured Ad-HBV infected hepatocytes in high- and low-glucose culture condition and examined impact of glucose on viral biosynthesis. We found that the reduction of glucose support in the culture medium significantly decreased extracellular secretions of HBsAg and HBeAg (Fig 7A), without affecting general cell viability (Fig 7B). In addition, reduction of intracellular LHBS, but not HBcAg, was detected under low glucose condition (Fig 7C). Accordingly, we suspected that synthesis of HBsAg was more sensitive to the glucose supply than that of HBcAg.

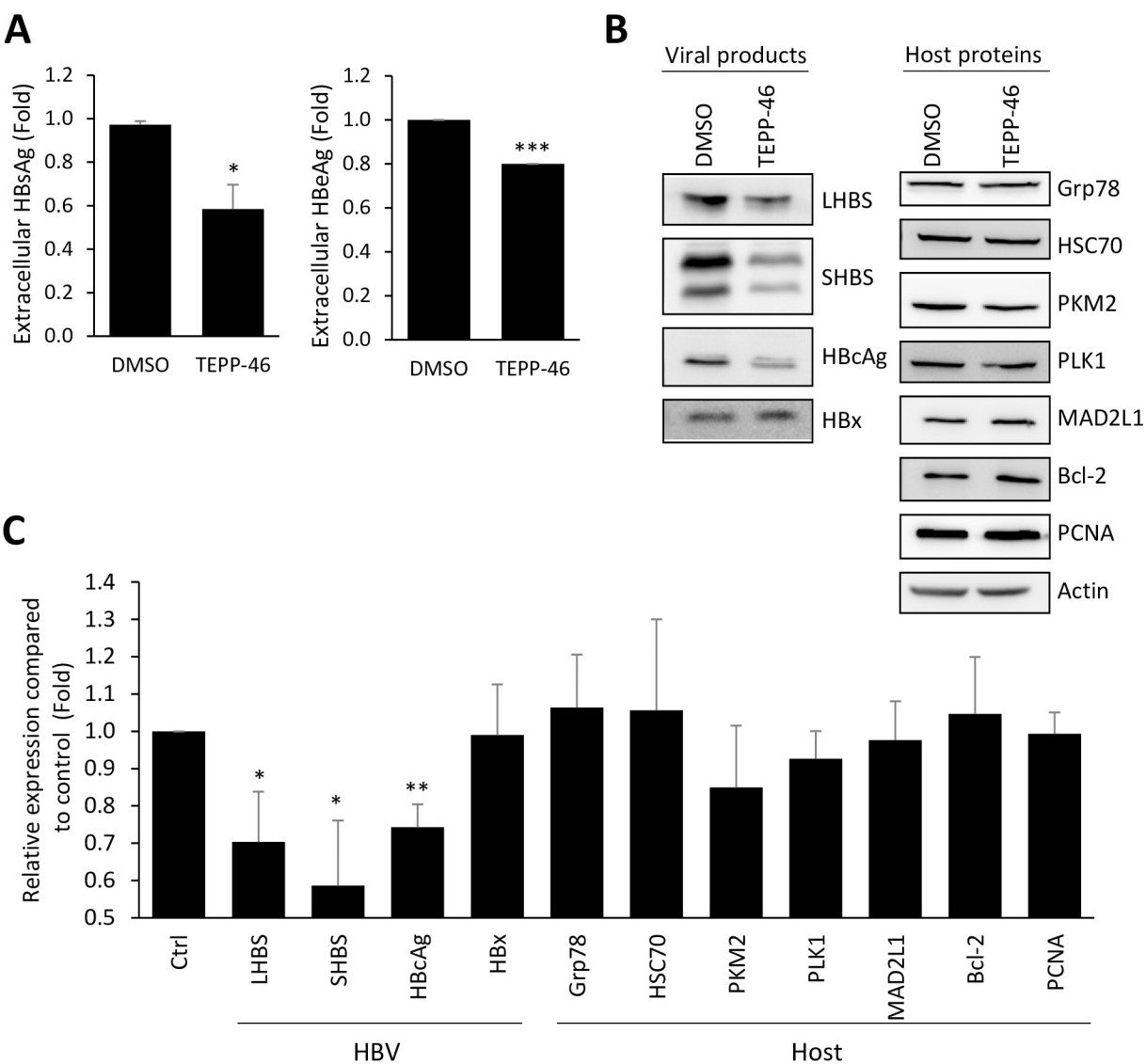

**Fig 5. TEPP-46 reduces protein synthesis of HBV in HuH-7 cells.** HuH-7 cells were transfected with pHBV3.6 for 24 hours and then treated with DMSO or 20 μM TEPP-46 for additional 24 hours. (A) The culture media of DMSO or TEPP-46 treatment were collected and subjected for measurement of HBsAg and HBeAg level. (B) Protein expressions of HBV viral products (left panel) and host proteins (right panel) upon treatment of DMSO or TEPP-46 were shown in western blotting. (C) Quantitative results of viral and host protein expressions were shown, with actin as an internal control. All quantitative results were a summary of three repeats and data were displayed as mean ± standard error. *p<0.05, **p<0.01, and ***p<0.001 were calculated using Student's t test.

We next investigated if HBV protein synthesis is affected by treatment of 2-Deoxy-D-Glucose (2-DG), a glucose analogue that can act to competitively inhibit the production of glucose-6-phosphate from glucose by phosphoglucoisomerase, thereby interrupting glycolysis. We found that the treatment of 2-DG reduced glucose consumption and lactate production, in a dose-dependent manner (Fig 8A). Coordinately, extracellular secretions of HBsAg and HBeAg, and intracellular levels of LHBS and HBcAg were reduced by 2-DG (Fig 8B and 8C). Taken together, our data indicated that major viral product synthesis, especially HBsAg and HBcAg, can be restrained by modulating glucose metabolism.

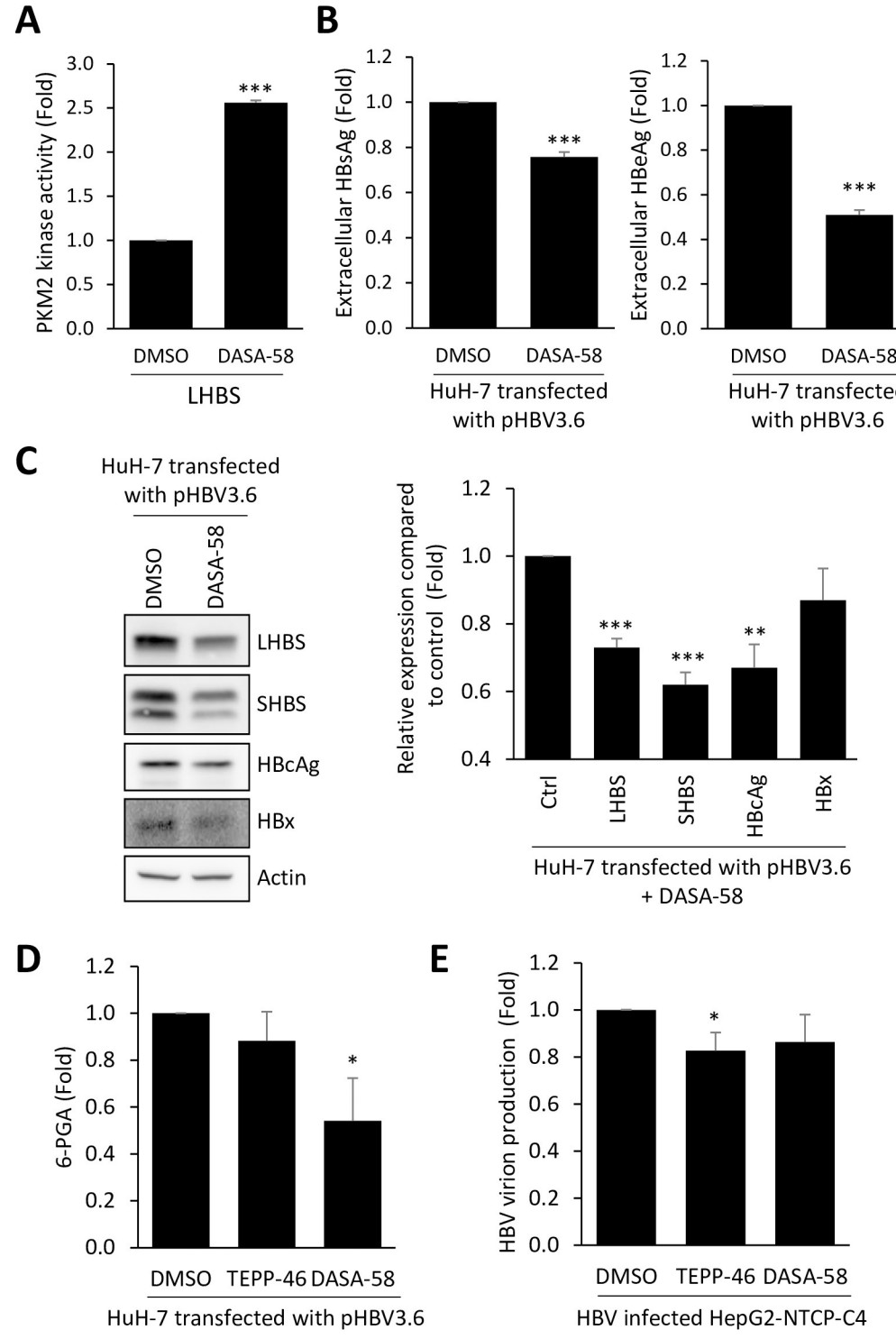

**Fig 6. Activators of PKM2 diminished HBV reproduction.** (A) LHBS-positive immortalized hepatocytes were treated with DMSO or 50 μM DASA-58 for 2 days. Total cell lysates were collected for measuring PKM2 activity. Relative PKM2 kinase activity of LHBS-positive cells was shown after normalization with that of DMSO treatment cells. (B) HuH-7 cells were transfected with pHBV3.6 for 24 hours and then treated with DMSO or 50 μM DASA-58 for additional 24 hours. Production of HBsAg and HBeAg in the culture supernatants were measured. Relative extracellular HBsAg and HBeAg were shown with DMSO-treated group as an internal control. (C) HuH-7 cells were transfected with pHBV3.6 for 24 hours and then treated with DMSO or 50 μM DASA-58 for additional 24 hours. Representative western blots of HBV viral products were shown on the left panels. The right panel showed changes of

indicated viral products after DASA-58 treatment in comparison to DMSO-treated cells as the control. (D) Intracellular abundance of 6-PGA was measured in pHBV3.6-transfected HuH-7 cells following treatments of DMSO, 20 μM TEPP-46 or 50 μM DASA-58 for 24 hours. Quantitative results of 6-PGA relative abundance were shown. (E) The production of virions in HBV-infected HepG2-NTCP-C4 cells, with or without treatments of chemical PKM2 activators were measured by detecting HBV DNA in the culture supernatant using quantitative PCR. All quantitative results were a summary of three repeats and data were displayed as mean ± standard error. *$p<0.05$, **$p<0.01$, and ***$p<0.001$ were calculated using Student's t test.

## Activators of PKM2 suppress LHBS-mediated oncogenesis

Intrahepatic LHBS has been reported as a priming factor for the development of HCC [25]. The expression of LHBS induced cytokinesis failure and consequent aneuploidy via induction of DNA damage and polo-like kinase 1 (PLK1)-mediated G2/M checkpoint failure in hepatocytes [12]. Whether aerobic glycolysis involved in LHBS-mediated oncogenesis is yet to be determined. Here we showed that LHBS induced anchorage-independent growth of immortalized hepatocytes on the 3-dimensional condition of soft agar plates. In contrast, the control

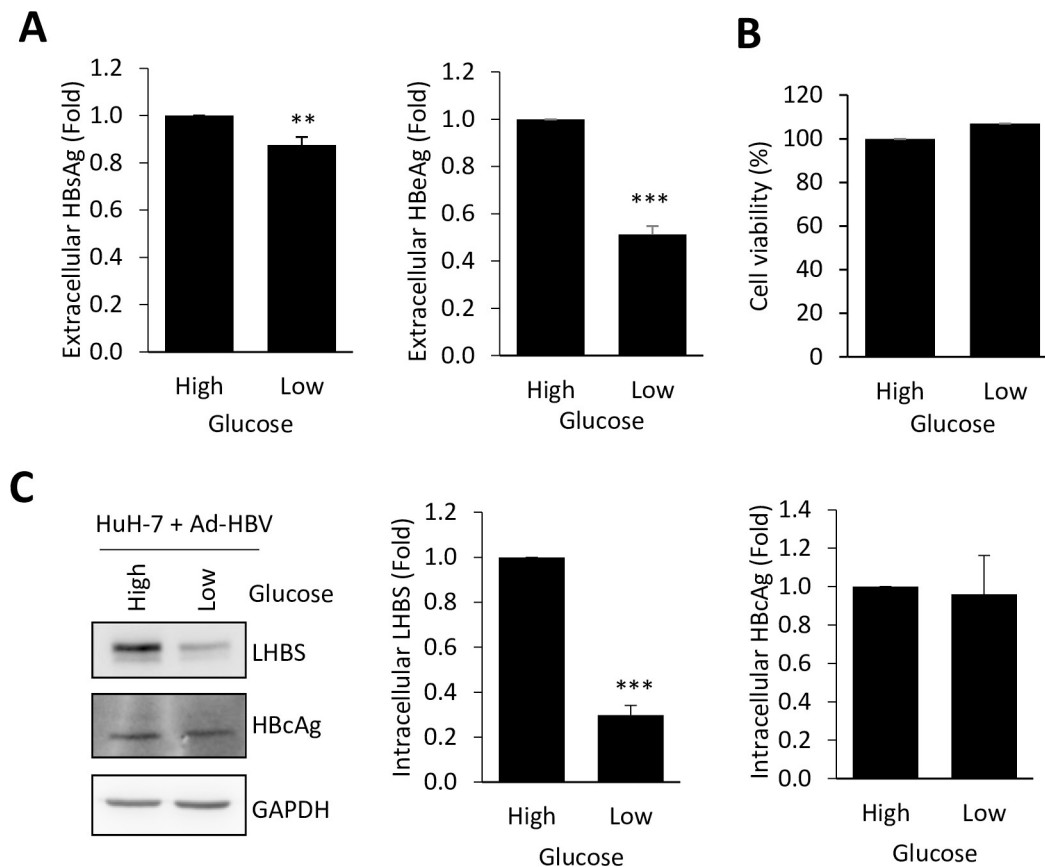

**Fig 7. Reduction of glucose support suppresses protein synthesis of HBV.** (A) HuH-7 cells were infected with Ad-HBV-WT at 200 MOI for 24 hours and then changed the medium containing high (4.5g/L) or low (1g/L) glucose for additional 24 hours. Extracellular levels of HBsAg and HBeAg were measured by the Cobas assay. The bar charts showed quantitative results after normalization with results obtained from the high-glucose condition. (B) General cell viabilities measured in high- and low-glucose conditions were shown. (C) Representative blots of LHBS, HBcAg, and GAPDH in Ad-HBV infected HuH-7 cells under different glucose supply were shown. All quantitative results were a summary of at least three repeats and data were displayed as mean ± standard error. **$p<0.01$, and ***$p<0.001$ were calculated using Student's t test.

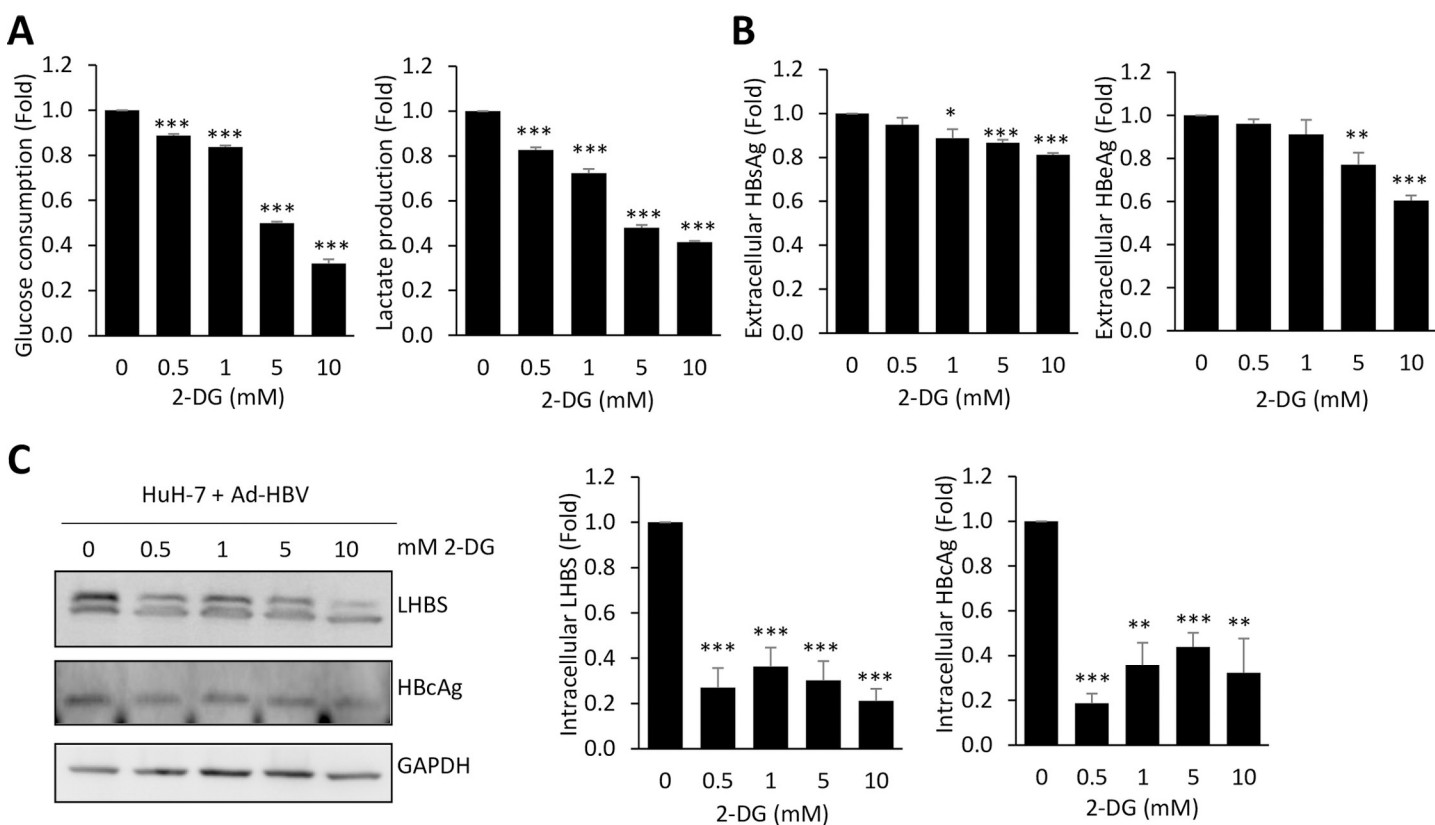

**Fig 8. HBV protein synthesis is suppressed by 2-DG.** HuH-7 cells were infected with Ad-HBV-WT for 24 hours and then treated with 0, 0.5, 1, 5 and 10mM of 2-DG for another 24 hours. The culture media were collected for measuring glucose consumption and lactate production (A), and the secretion of HBsAg and HBeAg (B). (C) Total protein lysates of each treatment were collected for the detection of intracellular LHBS and HBcAg by Western blotting. Representative blots were shown on the left panels. Quantitative results showing relative expressions of LHBS and HBcAg were shown, using non-treated groups (0 mM) as an internal control. All quantitative results were a summary of at least three repeats and data were displayed as mean ± standard error. *p<0.05, ** p<0.01, and ***p<0.001 were calculated using Student's t test.

lines failed to grow on soft agar plates (Fig 9A). We showed that TEPP-46 restored PKM2 activity (Fig 9B) without affecting general cell viability under traditional 2-dimensional culture (Fig 9C). However, cell growth on soft agar plates was largely diminished by TEPP-46 (Fig 9D), as well as DASA-58 (Fig 9E). Taken together, treatments of PKM2 activators could largely suppress LHBS-mediated anchorage-independent growth in hepatocytes. These data therefore provided a strong evidence to support the role of PKM2 as a negative regulator of HBV-mediate hepatocarcinogenesis.

## Discussion

Viral hijacking of cellular metabolism has been noted for over half a century. Pushing the metabolic flux into biosynthetic pathways is the key to support *de novo* synthesis of viral building blocks. The mechanisms and consequences of virus-induced metabolic switch, or reprogramming, have only begun to be studied in detail over the past decade [23]. This study adds a new model mechanism to illustrate how glucose metabolism is switched to aerobic glycolysis through the interaction between a key metabolic enzyme and a viral product, in this case, PKM2 and HBV LHBS. Noted that metabolic phenotypes conferred by viruses often imitate metabolic changes in cancer cells. As such, increased glucose consumption and lactate production are two major phenotypes of Warburg effect, a well-known cancer metabolism that

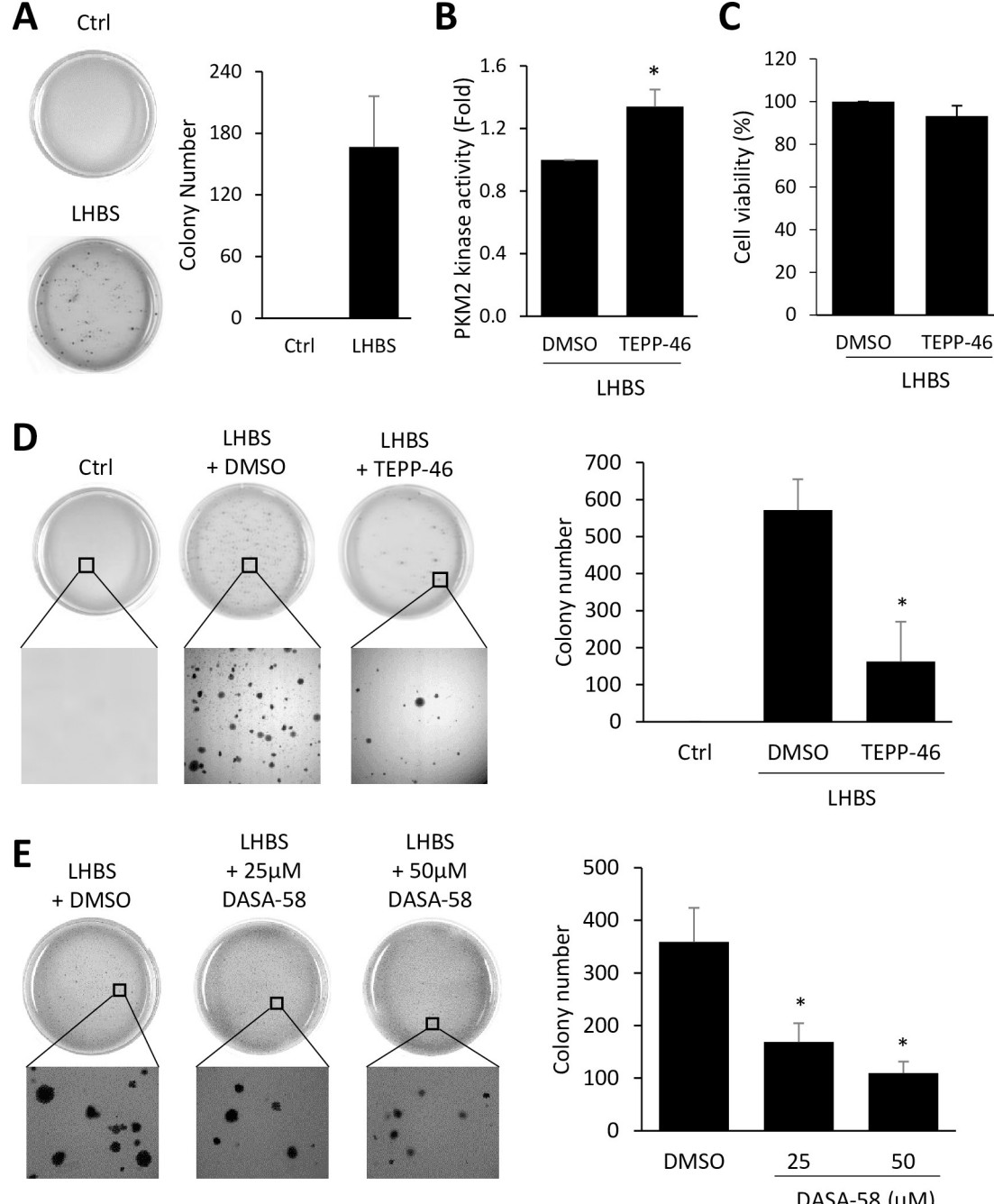

**Fig 9. PKM2 activators suppressed LHBS-mediated oncogenesis on immortalized hepatocytes.** (A) Colony formation assay was performed with control and LHBS-positive immortalized hepatocytes for 3 weeks in soft agar plates. Representative plate images are shown. No colony formation was detected in control cells. Quantitation of colony numbers were provided in the right panel. (B) LHBS-positive hepatocytes were treated with DMSO or 20 μM TEPP-46 for 24 hours and cell lysates were harvested for measuring PKM2 activity. (C) Cell viability was measured in LHBS-positive hepatocytes 48 hours after treatments with DMSO or 20 μM TEPP-46. (D) Representative colony formation images of control, and LHBS-positive hepatocytes treated with DMSO or 20 μM TEPP-46 were shown. Quantitation results of colony numbers were shown on the right panel. (E) Representative colony formation images of control, and LHBS-positive hepatocytes treated with DMSO or 50 μM DASA-58 were shown. Quantitation results of colony numbers were shown on the right panel. All quantitative results were a summary of three repeats and data were displayed as mean ± standard error. *$p < 0.05$ was calculated using Student's t test.

depicts increased glycolysis in the presence of oxygen. In cancer cells, such metabolic repro-
gramming is likely beneficial, as it allows rapid biosynthesis to support growth and prolifera-
tion. Aerobic glycolysis also enhances disruption of tissue architecture and immune cell
evasion in the tumor microenvironment [26]. As for viruses, the most obvious beneficial effect
comes from the flux into biosynthetic pathways, which support the *de novo* synthesis of viral
building blocks [23]. In this study, we find that HBV protein synthesis, especially HBsAg, relies
on the metabolic switch regulated by PKM2. In this study, several experimental models were
used to demonstrate the impact of HBV on the attenuation of PKM2 activity, including infec-
tious models of HepG2-NTCP-C4 and Ad-HBV, and pHBV3.6 transfection, as well as immor-
talized hepatocytes stably expressing LHBS. Among these experimental models, PKM2 activity
decreased the most in LHBS-positive immortalized hepatocytes than other conditions. This
can be explained by experimental variations among viral infections including HBV virions (as
in HepG2-NTCP-C4) and Ad-HBV (as in HuH-7 and HepG2), in compared with the transfec-
tion efficiency using pHBV3.6, and immortalized hepatocytes with whole cell population posi-
tive for LHBS. Interestingly, restoration of PKM2 activity only reduced expressions of HBV
viral products but not general host machinery (Fig 5). Here we show that expressions of 8 dif-
ferent host proteins, including mitotic and non-mitotic proteins, were not affected by TEPP-
46, whereas most viral products were reduced (Fig 5B). The impact of PKM2 activator on
HBV is restricted at the translational level, but not the transcription, as HBV RNA was not
affected upon PKM2 activation (S3 Fig). In addition, general stabilities of viral products were
not affected upon PKM2 activation (S2 Fig).

Why the protein synthesis of HBV is more sensitive to metabolic rewiring than the host
machinery is an interesting question to be clarified. As viral replication is a process involving
an extremely high turnover rate, a shortage in supply of amino acid precursors may respond
quickly to delay viral protein synthesis. In this regard, hijacking the metabolic supplies of mac-
romolecules from the host is a common strategy used by viruses for reproduction [23,27]. The
pentose phosphate pathway is a fundamental metabolism component that provides precursors
for nucleotide and amino acid biosynthesis, thus serving as ideal support to fuel viral repro-
duction. We suspect that HBV hijacks newly synthesized building blocks, probably coming
from the pentose phosphate pathway, for the demand of viral reproduction at high speed. This
notion explains why protein synthesis of HBV is more sensitive to metabolic rewiring than
that of general host proteins, as also supported by our observation that 6-PGA, a pentose phos-
phate pathway product, was reduced upon PKM2 activation (Fig 6D). In short, our results
imply that an efficient HBV reproduction relies on sufficient supports from the pentose phos-
phate pathway and aerobic glycolysis. Accordingly, targeting the host metabolic switch can be
a promising approach to control viral replication during chronic HBV infection.

In addition to supporting viral protein synthesis, increased glucose consumption in HBV-
infected cells also implies that these cells have a great need for energy, similar to highly prolif-
erating cells or cancer cells. Interestingly, general viability of LHBS cells were not affected by
TEPP-46 or DASA-58 when cultured in traditional two-dimensional (2D) cell culture, but
were largely suppressed when cultured in soft-agar plates (Fig 9). These results indicate that
aerobic glycolysis can be irrelevant under 2D culture, but is essential for cell adaptation to
three-dimensional (3D) culture condition. We suspect that cell growth in 3D soft agar can be a
stress condition for immortalized hepatocytes, as the parental immortalized hepatocytes grew
well in the 2D culture but failed to form colony formation in the soft agar (Fig 9A). Therefore,
the expression of LHBS was critical for the induction of anchorage-independent growth of
immortalized hepatocytes, on top of aerobic glycolysis. LHBS is a prominent viral oncoprotein
[12,28], and transgenic mice expression LHBS developed liver dysplastic changes and pro-
moted hepatocarcinogenesis [29]. Thus, we suggest that both PKM2-mediated metabolic

switch and the oncogenic property of LHBS contributed simultaneously to the cell adaptation and proliferation on the 3D culture. Accordingly, treatments with PKM2 activators may have two beneficial effects on patients with chronic HBV infection: to suppress viral protein synthesis and to prevent proliferation of virus-infected hepatocytes under 3D physiological condition. It has been shown that TEPP-46 has a good oral bioavailability *in vivo* and showed a promising effect in reducing tumor size in lung cancer xenograft tumors [24]. Accordingly, we suggest that TEPP-46 may be considered as a chemoprevention agent for patients with chronic HBV infection.

In this study, we also found that low glucose condition limited HBV protein synthesis in hepatocytes, implying that glucose support may affect viral reproduction. Notably, chronic hepatitis B patients with diabetes were shown to have 2.3-fold increased risk of developing HCC [30]. The link between serum glucose level and HBV replication is not clarified at this stage. It will be interesting to learn if controlling blood sugar may prevent HCC development in chronic hepatitis B patients with sustained diabetes. In this case, anti-diabetes agents may be applied to control HBV biosynthesis in patients with diabetes. On the other hand, we don't know whether anti-diabetic agents, such as metformin and alpha-glucosidase inhibitors, can be used in patients without diabetes and whether anti-diabetic agents not leading to hypoglycemia are still beneficial in such conditions. These questions are waiting for further investigations.

In summary, our data provide a mechanistic view to explain how metabolic switch is induced upon chronic HBV infection and thereby to support protein synthesis of HBV. This study suggests that rewiring the host metabolic switch may be applied to control viral reproduction and hepatocarcinogenesis in patients with chronic hepatitis B.

## Materials and methods

### Cell culture and transfection

The hTERT-immortalized hepatic progenitor cell line NeHepLxHT was obtained from the American Type Culture Collection (ATCC, Manassas, VA, USA). Stable cell lines expressing a SNAP tag (as a control) or SNAP-tagged LHBS were established from NeHepLxHT cells that have been described previously [16]. Both of the stable cell lines were cultured on type-I collagen dish in Dulbecco's modified of Eagle's medium/Ham's F-12 50/50 mix (DMEM-F12, Corning) containing L-glutamine and 15 mM HEPES and supplemented with 15% fetal bovine serum (FBS) (Biological industries), 100 IU penicillin, 100 μg/ml streptomycin (Corning), ITS premix (5 μg/ml insulin, 5 μg/ml human transferrin, and 5 ng/ml selenic acid; BD), 20 ng/ml epidermal growth factor (BD), and 100 nM dexamethasone (Sigma). HepAD38 [20] and HepG2-NTCP-C4 [18] cells were cultured in Dulbecco's modified of Eagle's medium/Ham's F-12 50/50 mix (DMEM-F12, Corning) containing L-glutamine and 15 mM HEPES and supplemented with 10% FBS (Biological industries), 100 IU penicillin, 100 μg/ml streptomycin (Corning), ITS premix (5 μg/ml insulin, 5 μg/ml human transferrin, and 5 ng/ml selenic acid; BD) and 400 μg/ml G418 (Sigma). HepAD38 cells were maintained with 1.5 μg/ml doxycycline when terminating HBV induction. 293T, HepG2 and HuH-7 cells were cultured in standard DMEM-based media. All the cell lines incubated at 37°C in a humidified atmosphere containing 5% $CO_2$. DNA transfection was performed with GeneJet In Vitro DNA Transfection Reagent (Ver. II) (SignaGen Laboratories) according to the manufacturer's instructions.

### HBV production and infection

The HBV used in this study was genotype D derived from HepAD38 cells as described [20]. In brief, culture supernatants of HepAD38 cells grown under doxycycline-free condition were

harvested every 2 days for 20 days. Collected supernatants were centrifuged at 3000 rpm for 10 minutes and filtered through 0.45μm filter. The HBV virions were purified by 5 ml sucrose cushion (20% sucrose, 100mM NaCl, 10mM Tris-HCl pH7.5 and 1mM EDTA pH8.0) ultra-centrifugation at 25000 rpm for 16 hours at 4˚C using SW 28 Ti Swinging-Bucket Rotor (Beckman Coulter). The precipitates were resuspended with serum free medium with 100-fold concentration. HBV infection on HepG2-NTCP-C4 cells was conducted as described [18]. To increase cell susceptibility, HepG2-NTCP-C4 cells were pretreated 2.5% DMSO for 2 days and then performed suspension infection. The cells were infected with HBV at a multiplicity of infection (MOI) of 1000 genome equivalent (GEq)/cell in the presence of 4% PEG8000 and 2.5% DMSO.

### Extraction of HBV DNA and quantitation

HBV DNA was extracted from the purified media using QIAamp MinElute Virus Spin kit (Qiagen) according to manufacturer's instruction. HBV DNA was quantified by real time PCR analysis using the primers 5'-CACCTCTGCCTAATCATC-3' and 5'-CGATACAGAGCT-GAGGCGGT-3'. The reactions were carried out using 2x SensiFAST SYBR Lo-ROX Mix (BIOLINE) and real time PCR was performed at 95˚C for 2 minutes and 40 cycles of 95˚C for 10 seconds and 60˚C for 30 seconds on AriaMx Real-Time PCR System (Agilent Technologies). The viral genome equivalent copies were calculated based on a standard curve generated with known copy numbers of plasmid (pCMV_HBV genotype D precore null, kindly provided by Dr. Pei-Jer Chen, National Taiwan University, Taiwan).

### Infection with Ad-HBV

Ad-HBV-KO and Ad-HBV-WT were the kind gifts from Dr. Li-Rung Huang (National Health Research Institutes, Taiwan) and produced according to a previous publication [22]. Specifically, Ad-HBV was produced by HEK293 cell transfection with adenovirus package plasmids. The HBV1.3 and HBV1.3 KO genomes were replaced into the E1 region of adeno-virus (Ad5ΔE1/E3) backbone plasmid. HBV1.3 KO contained premature stop codons in all HBV open reading frames. For the infection with Ad-HBV, HuH-7 and HepG2 cells were seeded in 6 cm dishes overnight and infected with Ad-HBV-KO or Ad-HBV-WT at indicated MOI of 0 to 200 GEq/cell for 2 days.

### Plasmids

The pHBV3.6 construct containing 1.2 copies of HBV genome was described previously [31]. For constructing different truncated surface antigens, LHBS, MHBS, SHBS and PreS of surface antigens were fused downstream of SNAP tag of a pSNAP-tag(m) vector (New England Bio-Labs) to generate SNAP-LHBS, SNAP-MHBS, SNAP-SHBS and SNAP-PreS, respectively. Plasmids encoding HA-tagged full length of PKM2, amino acids 110–531, 1–476, 1–421 and 1–366 were kindly provided by Dr. Wen-Ching Wang (National Tsing Hua University, Taiwan) as described previously [32]. The amino acids 367–476 of PKM2 was fused downstream of HA tag of pcDNA3.1-HA vector.

### Reagents

TEPP-46 (ML-265) and 2-deoxy-D-Glucose (2-DG) were purchased from Cayman Chemical. DASA-58 was purchased from SelleckChem. Dimethyl sulfoxide (DMSO, Sigma) was used as vehicle control. All inhibitor stocks were dissolved in DMSO to make the following stock concentrations: TEPP-46, 20 mM; 2-DG, 1 M; DASA-58, 50 mM.

## Antibodies

The following primary antibodies were used in this study: rabbit anti-PKM2 (CST 4053, Cell Signaling Technology), rabbit anti-HBsAg (ad/ay, PAB13969, Abnova), normal mouse IgG (sc-2025, Santa Cruz Biotechnology), normal rabbit IgG (sc-2027, Santa Cruz Biotechnology), rabbit anti-HBcAg (B0586, DAKO), mouse anti-HA-tag (sc-7392, Santa Cruz Biotechnology), mouse anti-HSC70 (sc-7298, Santa Cruz Biotechnology), rabbit anti-SNAP-tag (P9310, New England BioLabs), rabbit anti-beta-tubulin (NB600-936, Novus Biologicals), mouse anti-beta-actin (NB600-501, Novus Biologicals), rabbit anti-GAPDH (GTX100118, GeneTex), rabbit anti-Grp78 (GTX113340, GeneTex), rabbit anti-PLK1 (GTX104302, GeneTex), rabbit anti-MAD2L1 (GTX104680, GeneTex), rabbit anti-Bcl-2 (GTX100064, GeneTex), and rabbit anti-PCNA (GTX100539, GeneTex). Mouse anti-LHBS/PreS1 (7H11), mouse anti-SHBS (86H6), and mouse anti-HBx (20F3) were kindly provided by Professor Ning-Shao Xia (Xiamen University, China) [33]. The following secondary antibodies were used in this study: peroxidase-conjugated AffiniPure goat anti-rabbit IgG (H+L, 111-035-003, Jackson ImmunoResearch), peroxidase-conjugated AffiniPure goat anti-mouse IgG (H+L, 115-035-003, Jackson ImmunoResearch), Alexa Fluor 488 donkey anti-rabbit IgG (H+L, A21206, Invitrogen) and Alexa Fluor 647 donkey anti-mouse IgG (H+L, A31571, Invitrogen).

## Western blot analysis

Cells were lysed in RIPA buffer (50 mM Tris pH7.5, 150 mM Sodium chloride, 1% NP40, 0.5% Sodium deoxycholate, and 0.1% SDS) containing protease inhibitor cocktail (Roche). Total cell lysates were harvested by collecting supernatant after centrifugation at 13000 rpm at 4°C for 15 minutes. Harvested cell lysates were mixed 1:1 with 2x laemmli buffer (0.04% SDS, 20% glycerol, 120mM Tris pH6.8) containing β-ME and denatured at 100°C for 15 minutes before being subjected to SDS-PAGE and subsequently transferred to a PVDF membrane (Millipore). Membranes were incubated overnight at 4°C with indicated primary antibodies diluted in 5% non-fat milk and $PBST_{0.05}$, washed $3 \times 5$ minutes in $PBST_{0.05}$ and incubated with appropriate HRP conjugated secondary antibodies at room temperature for 1 hour. The membranes were incubated with Western Lightning Plus ECL (PerkinElmer. Inc) reagent and scanned using the ImageQuant LAS 4000 digital imaging system (GE Healthcare). Rabbit anti-beta-tubulin, mouse anti-beta-actin and rabbit anti-GAPDH were used as internal controls to normalize protein loading in a western blot.

## Immunoprecipitation

Cells were lysed with RIPA buffer (50 mM Tris pH7.5, 150 mM Sodium chloride, 1 mM EGTA, 0.5% NP40, 0.1% Sodium deoxycholate) containing one protease inhibitor cocktail and 1mM DTT. Took 40 μl protein G Mag Sepharose (GE Healthcare) and incubated with antibody at 4°C at least 30 minutes. The cell lysates were collected and diluted with RIPA buffer to final concentration of 2 μg/μl. Took part of the cell lysate into a new tube and added equal volume of 2x laemmli buffer, boiled at 100°C for 15 minutes (Input). The rest of cell lysates were added into antibody-Mag sepharose mixture and rotated at 4°C overnight. Beads were washed thrice with ice-cold RIPA buffer, resuspended in 2x laemmli buffer and then denatured by boiling at 100°C for 15 minutes (IP product). Samples were subjected to western blot analysis.

## Proximity ligation assay

Duolink In Situ Red Starter Kit Mouse/Rabbit (DUO92101) was purchased from Sigma-Aldrich and the assay followed Duolink PLA Fluorescence Protocol provided from the

manufacturer. In brief, cells were seeded on collagen-coated coverslips, washed with PBS, and then fixed with 4% formaldehyde/PBS at room temperature for 10 minutes. The coverslips were treated with Duolink blocking solution at 37°C for 1 hour then incubated with mouse anti-LHBS/PreS1 (7H11) and rabbit anti-PKM2 antibodies for another 1 hour. The coverslips were washed three times with PBS, and applied PLA probes (anti-rabbit PLUS and anti-mouse MINUS) at 37°C for 1 hour and continued with probe ligation and amplification according to manufacturer's instruction. After signal amplification of PLA signals, the slides were incubated with donkey anti-mouse conjugated with AlexaFluor-647 and donkey anti-rabbit conjugated with AlexaFluor-488 for 1 hour, washed three times in PBS-0.05% Tween20, and mounted with DAPI-containing mounting medium. The fluorescence cell images were taken by Leica DMI6000 inverted microscope (Leica Microsystems) and an Andor Luca R EMCCD camera (Andor Technology). All images were acquired and processed with MetaMorph software (Molecular Devices).

### PKM2 kinase activity assay

The PKM2 kinase assay was performed following the protocol of Pyruvate Kinase Activity Colorimetric/Fluorometric Assay Kit (BioVision). In brief, cells were extracted with 4 volumes of assay buffer then centrifuged to collect supernatant containing PKM2. Total protein lysates were diluted to the same concentration in the assay buffer and 50 μl of diluted samples were loaded onto 96 well plates. For each reaction, 50 μl reaction buffer was added for each well and measured OD 570 nm at 0 and 20 min. Protein kinase activity of PKM2 was calculated against the standard substrate provided in the kit.

### Measuring glucose consumption and lactate production

Cells were cultured on 96-well plate for 3 days and the supernatant was collected. Glucose consumption was analyzed using glucose assay kit (Eton Bioscience), according to manufacturer's protocol. Lactate production was analyzed using lactate assay kit (Eton Bioscience), followed the manufacturer's protocol.

### PKM2 oligomerization assay

Performing the cross-linking reaction, the cell lysate was equally separated in two tubes and treated with equal volume of PBS (as a control) or 0.1% glutaraldehyde for 20 minutes at 25°C The reaction was stopped by adding Tris buffer (pH8.0) to a final concentration of 50mM Tris. Added 2x laemmli buffer and boiled at 100°C for 15 minutes, followed by western blotting.

### Glycerol-gradient centrifugation for PKM2 oligomer separation

Cells were freshly harvested and lysed in 1x TBS lysis buffer (50 mM Tris pH7.5, 150 mM Sodium chloride and 0.1% NP40) containing protease inhibitor cocktail (Roche). Cell lysates were centrifuged at 13000 rpm at 4°C for 15 minutes and collected the supernatants. A total of 150 μg proteins was loaded on the top of a 15–35% glycerol gradient (using 50 mM Tris pH7.5 and 150 mM Sodium chloride) and ultracentrifuged at 50000 rpm for 16 h at 4°C using SW 60 Ti Swinging-Bucket Rotor (Beckman Coulter). After centrifugation, a total of 36 fractions were collected for each density gradient sample and subjected to Western blot analysis for PKM2 oligomerization.

## Measuring cell viability

Cells ($1.5x10^3$/well) were seeded in 96 well plate overnight and added drugs for 3 days. Removed medium and added 100 μl (10 μl CCK-8 + 90 μl medium) diluted CCK-8 in each well. Added CCK-8-contained medium into extra well without cells as background control. Incubated at 37°C for 2 hours. Cell viability was measured at OD450/655 nm.

## Colony formation assay

NeHep-SNAP-LHBS cells were seeded in 6 cm dish and drug treatment of adding DMSO (control), 20 μM TEPP-46, 25 μM or 50 μM DASA-58 for 2 days. Each 6 cm dish, prepared the bottom agar layer with 1 ml mixture containing $ddH_2O$, 2x complete medium, and 2.4% gel (1:2:1). Kept at room temperature becoming solid. Next, trypsinized cells into single cell suspension and counted cell numbers to adjust to $5x10^4$ cells/ml with normal medium. Prepared the top agar layer with 2 ml mixture containing $ddH_2O$, 2x complete medium, 2.4% gel (2:1:1), and $2x10^3$ cells in 1 ml normal medium. After top agar gel became solid, added 1.5ml normal medium to provide nutrients. Incubated at 37°C for 3 weeks and added 1.5 ml complete medium on top of the gel once per week. Dyed the colonies using 10% trypan blue and washed with $ddH_2O$, followed by counting colony numbers.

## Measure extracellular HBsAg and HBeAg

The culture supernatant was collected and centrifuged at 13000 rpm for 5 minutes to remove cell debris. HBsAg and HBeAg were measured by Cobas assay following manufacturer's instruction. The following kits were used in this study: Elecsys HBsAg II (04687787, cobas, Roche) and Elecsys HBeAg (11820583, cobas, Roche) by cobas e 411 analyzer (Roche).

## 6-Phosphogluconic Acid (6-PGA) Assay

The assay followed the protocol of 6-Phosphogluconic Acid (6-PGA) Assay Kit (Colorimetric) (BioVision). In brief, cells were extracted with the assay buffer then centrifuged to collect supernatant. Total protein lysates were diluted to the same concentration in the assay buffer and 50 μl of diluted samples were loaded onto 96 well plates. For each reaction, 50 μl reaction buffer was added for each well. Incubated at 37°C for 1 hour and measured OD 450 nm. The concentration of 6-PGA was calculated against the standard substrate provided in the kit.

## Quantification and statistical analysis

All experiments were repeated independently at least three times. Protein quantitations were done by detecting signal intensities of Western blots using ImageQuant LAS 4000 digital imaging system (GE Healthcare) and quantified using the accompanying ImageQuant TL software (GE Healthcare). Subtraction of background signals was done independently for each individual blot. All quantifications were performed within the linear range of the instruments in the absence of signal saturation. To control for possible variations in sample loading, protein signals were normalized to internal control signals of GAPDH, tubulin, or actin, as indicated, then calculated the fold changes between experimental and control groups. For PKM2 oligomerization analysis, the signal detected for the dimer fractions was normalized with the total PKM2 detected as monomer in the non-crosslinking counterpart. Statistical analysis was performed using Student's t test. $p < 0.05$ is considered significant. All data were displayed as mean ± standard error.

## Supporting information

**S1 Fig. Expressions of MHBS and SHBS reduced PKM2 activity.** (A) HuH-7 cells were transiently transfected with pHBV3.6 wild type (WT), start codon mutant on PreS1 (ΔL) and double start codon mutant on PreS1/S2 (ΔL/M) for 48 hours. Expressions of LHBS and SHBS were detected by Western blotting. (B) PKM2 kinase activity was measured on transfected hepatocytes at 48 hours post-transfection. The quantitative result was a summary of five repeats and data were displayed as mean ± standard error. n.s., not significant. * p<0.05 and ** p<0.01 were calculated using Student's t test.
(TIF)

**S2 Fig. Viral protein stabilities are not affected by PKM2 activators.** HuH-7 cells were transfected with pHBV3.6 for 24 hours and then treated simultaneously with 100 μg/ml cycloheximide, DMSO, 20 μM TEPP-46 or 50 μM DASA-58, as indicated. Total cell lysates were collected at indicated time points and subjected to western blotting using antibodies against LHBS/PreS1, SHBS, HBcAg, HBx, and GAPDH. Representative blots are shown.
(TIF)

**S3 Fig. The impact of PKM2 activation on viral transcription.** HuH-7 cells were transfected with pHBV3.6 for 24 hours and then treated with DMSO, 20 μM TEPP-46 or 50 μM DASA-58 for additional 24 hours. Total cell RNA was extracted and subjected to reverse transcription and quantitative PCR using primers specific for SHBS, PreS1 and HBcAg coding sequences. The quantitative results were a summary of three independent repeats and data were displayed as mean ± standard error.
(TIF)

**S1 Table. List of interacting proteins of LHBS identified by mass spectrometry.**
(XLSX)

**S1 Text. Supporting materials and methods.**
(DOCX)

## Acknowledgments

We thank Dr. Daniel KY Wu for project support, Dr. Christoph Seeger (Fox Chase Cancer Center at Temple University Hospital, United States) for HepAD38 cell line, Dr. Pei-Jer Chen (Hepatitis Research Center, National Taiwan University Hospital, Taiwan) for HBV genotype D plasmid, Dr. Li-Rung Huang (National Health Research Institutes, Taiwan) for providing Ad-HBV, Dr. Wen-Ching Wang (Institute of Molecular and Cellular Biology, National Tsing Hua University, Taiwan) for providing HA-PKM2, Dr. Hui-Lin Wu (Hepatitis Research Center, National Taiwan University Hospital, Taiwan) and Mr. Cheng-Yen Chung (National Taiwan University, Taiwan) for technical assistance in the HBV production and infection.

## Author Contributions

**Conceptualization:** Yi-Hsuan Wu, Bor-Sen Chen, Lily Hui-Ching Wang.

**Data curation:** Tian-Neng Li.

**Formal analysis:** Yi Yang, Ching-Hung Chen, Kuan-Ju Liao.

**Funding acquisition:** Bor-Sen Chen, Lily Hui-Ching Wang.

**Investigation:** Yi-Hsuan Wu, Yi Yang, Ching-Hung Chen, Chia-Jen Hsiao, Tian-Neng Li, Lily Hui-Ching Wang.

**Methodology:** Yi-Hsuan Wu, Kuan-Ju Liao, Koichi Watashi.

**Resources:** Chia-Jen Hsiao, Koichi Watashi, Bor-Sen Chen.

**Writing – original draft:** Yi-Hsuan Wu, Lily Hui-Ching Wang.

**Writing – review & editing:** Lily Hui-Ching Wang.

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
