## [Decision Letter · Decision Letter 0]

24 Aug 2020

Dear Dr. Wang,

Thank you very much for submitting your manuscript "Aerobic glycolysis supports hepatitis B virus biosynthesis through interaction between viral surface antigen and pyruvate kinase isoform M2" for consideration at PLOS Pathogens. As with all papers reviewed by the journal, your manuscript was reviewed by members of the editorial board and by several independent reviewers. Although the reviewers recognized the interesting relationship between aerobic glycolysis and HBV, they also raised some serious issues. These issues include a weak 13% decrease of PKM2 activity in Ad-HBV-infected Huh7 cells, and the lack of statistical and quantitation analyses. Most notably, all three reviewers emphasized the need to use HBV infection systems to support your claims. These issues must be addressed in your revised manuscript for consideration of publication. 

We cannot make any decision about publication until we have seen the revised manuscript and your response to the reviewers' comments. Your revised manuscript is also likely to be sent to reviewers for further evaluation.

Sincerely,

Aleem Siddiqui, Ph.D.

Associate Editor

PLOS Pathogens

Jing-hsiung James Ou

Section Editor

PLOS Pathogens

Kasturi Haldar

Editor-in-Chief

PLOS Pathogens

orcid.org/0000-0001-5065-158X

Michael Malim

Editor-in-Chief

PLOS Pathogens

orcid.org/0000-0002-7699-2064

Reviewer's Responses to Questions

**Part I - Summary**

Reviewer #1: This is a well written manuscript by Y-H Wu and colleagues.

Introduction and Discussion sections are particularly informative.

The initial hypothesis that HBV governs intracellular biosynthesis in infected cells to its advantage is interesting because it may help understand better the biology of HBV including the process of oncogenesis. For this latter reason, the authors chose to select the L HBV env protein for the viral target of choice.

An affinity purification experiment was first conducted to identify L-binding partners using a L-expressing cell line previously developed in the laboratory. Unfortunately, there are no details provided on either the level of L expression in this cell line, the pulldown experiment and the identified interactants other than pyruvate kinase isoform M2 (PKM2).

Was PKM2 the major binder ? Because the initial pulldown was carried out in a somewhat artificial system (a tagged-L protein overexpressed in a non-hepatocyte cell-line), PKM2-L binding should be confirmed in HBV infected cells (PHH or NTCP-expressing cell lines).

Experiments reported in Fig.2 show that the C-terminus of PKM2 binds to the S domain of L HBV env. This suggests that S env protein would probably be the best binder to PKM2. Furthermore, S HBV env is, by far, the most abundant HBV protein in infected cells and should have logically been tested.

It is unclear to this reviewer whether the SNAP-L protein used as a bait in the pulldown experiments is myristoylated, and why the profiles of L, M and S proteins in western blot analysis (Fig.2B and D) are not conventional, for both size and glycosylation state. These observations are further indications that experiments should indeed be conducted in the context of an in vitro infection of HBV susceptible cells.

As the PKM2 protein activity is correlated to its degree of polymerization, the authors sought to evaluate the effect of L on PKM2 dimer and tetramer formation (Fig.3 and 4). The results presented in Fig.3B, 4B and 4E are not convincing, for lacking precise quantification and for ignoring that the levels of monomers appear to vary in parallel with dimers and tetramers Fig.4B and 4E.

The authors used TEPP-46 and DASA-58 drugs as known activator of PKM2 to treated HBV expressing Huh-7 cells. The experiment is not precisely described, but the results presented in Fig.6 show only a modest effect of the treatment on extra-cellular HBsAg (SVPs essentially) and HBeAg.

Overall the experiments reported in this manuscript lack information on statistical analysis and precise quantification. Measurement of western blot signals by digital imaging systems is not sufficiently precise.

Furthermore, it is not demonstrated that biosynthesis of HBV proteins, per se, is affected by aerobic glycolysis. What about protein degradation? And what about HBV DNA replication and virions production ?

The conclusion of the study that L-HBV env protein would induce a metabolic switch (increase of glucose consumption and lactate production) to favor virus production would require that a study be conducted on in vitro infected cells, using more precise quantification tools. The statement in the Author Summary paragraph: “ We show that metabolic switch not only favors biosynthesis of HBV but also provokes hepatocarcinogenesis” is not supported by the data.

Reviewer #2: The authors used an affinity purification screen to identify host factors that interact with The HBV large surface antigen (LHBS or L). This effort lead to the identification of the cellular pyruvate kinase isoform M2 (PKM2), a key regulator of glucose metabolism, as a L binding partner. They then showed that the expression of LHBS affected oligomerization of PKM2 in hepatocytes, and increased glucose consumption and lactate production, i.e., aerobic glycolysis. Restoring PKM2 activity by chemical activators, TEPP-46 or DASA-58, reduced the levels of viral surface and core antigens. Reduction of glycolysis by culturing cells in low-glucose condition or treatment with 2-deoxyglucose also decreased viral surface antigen levels. PKM2 activation by TEPP-46 suppressed proliferation of LHBS-positive cells in 3-dimensional agarose culture, without showing any

effect on the traditional 2-dimensional cell culture. The authors concluded that HBV-induced metabolic switch may support biosynthesis of HBV in hepatocytes. In addition, aerobic glycolysis is likely essential for LHBS mediated oncogenesis.

Whereas some of the authors’ results are tantalizing and potentially significant in understanding HBV-host interactions in HBV replication and pathogenesis, the results are subject to alternative interpretations and more rigorous studies in more physiologically relevant systems are needed to verify the observations.

Reviewer #3: Yi-Hsuan Wu and colleagues report in this manuscript that HBV large envelope protein LHBs interacts with pyruvate kinase isoform M2 (PKM2) and inhibits its enzymatic activity, which consequentially induces metabolic switch from oxidative phosphorylation to aerobic glycolysis, with increased glucose consumption and lactate production in hepatocytes. The authors claimed that the virus-induced metabolic switch not only favors the replication of HBV but also provokes hepatocarcinogenesis. Overall, the virus-induced metabolic reprogram is an important, but under-investigated area in HBV pathogenesis. The authors made some interesting findings, which require further mechanistic analysis and validation in HBV infected human hepatocytes.

**Part II – Major Issues: Key Experiments Required for Acceptance**

Reviewer #1: Pyruvate kinase isoform M2 (PKM2) a key regulator of glucose metabolism was identified. PKM2-L HBV env binding should be confirmed in HBV infected cells.

Because the S domain of L HBV env has been identified as responsible for binding to PKM2, S env protein should be tested.

Overall the experiments lack precise quantification and information on statistical analysis.

Quantification of western blot signals by digital imaging systems is sufficiently precise.

It is claimed that biosynthesis of HBV proteins, per se, is affected by aerobic glycolysis. Protein stability should be evaluated.

There is no analysis of HBV DNA and virions levels.

Reviewer #2: 1. Since the authors mapped the interaction domain on L to the S domain, would expression of S alone show the same effects on the cells as L? Is it possible the effects observed with overexpression of L was due to indirect consequences of L retaining in the cell, e.g., ER stress response?

2. Is it possible to further map the interacting region within the S domain? As S is normally rapidly secreted from the cell, S expression may not lead to high levels of intracellular S protein. However, it may be possible to express the interacting from S in the cell and see if it would have the same effects as L expression.

3. As HBV infection systems are now readily available, e.g., HepG2-NTCP or better, primary human hepatocytes, the authors need to verify the effects of L overexpression using these more authentic infection systems.

4. What are the effects of HBV on PKM2, and PKM2 on HBV, in the 3-dimensional agarose culture vs. the traditional 2-dimensional cell culture? Can any of those effects be correlated with the effects on cell proliferation?

Reviewer #3: 1. It is not clear whether N-terminal SNAP tag interferes with the membrane topology of LHBs, MHBs and SHBs proteins? It appears that all the experiments presented in Figs 1 to 3 used SNAP-tagged HBV envelope proteins. Because SNAP tag is a 182 residues polypeptide and might interfere with the structure and function of HBV envelope proteins, it is important to demonstrate PKM2 interacts with native L, M and S proteins in the context of HBV replication.

2. Because all the three envelope proteins, L, M and S, interact with PKM2, does the expression of MHBs or SHBs also inhibit the enzymatic activity of PKM2? It is rather striking that expression of SNAP-LHBs alone reduced PKM2 activity by 86%, but infection of Huh7 cells with Ad-HBV-WT only inhibited PKM2 activity by 13%, suggesting a very weak effect in the context of HBV replication.

3. In evaluating the relationship between PKM2 activity and HBV replication (Figs. 5 to 8), only viral protein expression/secretion was measured. In order to obtain further mechanistic insights, HBV RNA and core DNA should also be determined. More importantly, the effects of HBV on cellular glycolysis and impact of glycolysis on HBV replication should be further validated in HBV infected primary human hepatocytes, beacuse the glycolysis pathway of hepatoma cells (HepG2 and Huh7) may be altered.

4. If LHBs interaction with PKM2 underlines HBV oncogenic activity, it can be anticipated that MHBs and SHBs should have a similar effect? Is this true?

**Part III – Minor Issues: Editorial and Data Presentation Modifications**

Reviewer #1: (No Response)

Reviewer #2: 1. Fig. 1B: Images from a single cell are not acceptable! Quantitative results are needed.

2. Line 336: Ad-HBV-KO needs to be described more clearly.

3. Line 358-359: These Abs (for L, S, X) should be described in more details, or if published, relevant references need to be cited.

4. Line 390: More details are needed for the PLA assay - probe sequences, Abs used, assay conditions, etc.

Reviewer #3: No.

PLOS authors have the option to publish the peer review history of their article (what does this mean?). If published, this will include your full peer review and any attached files.

Reviewer #1: No

Reviewer #2: No

Reviewer #3: No
---

## [Decision Letter · Decision Letter 1]

15 Feb 2021

Dear Dr. Wang,

Thank you very much for submitting your manuscript "Aerobic glycolysis supports hepatitis B virus biosynthesis through interaction between viral surface antigen and pyruvate kinase isoform M2" for consideration at PLOS Pathogens. As with all papers reviewed by the journal, your manuscript was reviewed by members of the editorial board and by several independent reviewers. Based on the reviews, we are likely to accept this manuscript for publication, providing that you modify the manuscript according to the review recommendations.

Sincerely,

Aleem Siddiqui, Ph.D.

Associate Editor

PLOS Pathogens

Jing-hsiung James Ou

Section Editor

PLOS Pathogens

Kasturi Haldar

Editor-in-Chief

PLOS Pathogens

orcid.org/0000-0001-5065-158X

Michael Malim

Editor-in-Chief

PLOS Pathogens

orcid.org/0000-0002-7699-2064

Reviewer Comments (if any, and for reference):

Reviewer's Responses to Questions

**Part I - Summary**

Reviewer #1: In the revised version of the manuscript, the authors have satisfactorily answered the main questions of this reviewer. They have performed a considerable number of experiments to obtain a better quantification in their study, and have modified the text accordingly.

Reviewer #2: The authors have adequately addressed my previous concerns.

Reviewer #3: The authors made great efforts to address my concerns on the previous version. The manuscript is significantly improved. The following minor points are intend to help further improve the manuscript.

1. The word “viral biosynthesis” or “de novo biosynthesis” does not specifically refer to any specific viral replication step or biosynthesis event. As your results seem to indicate that activation of PMK2 selectively inhibit the synthesis of viral envelope proteins, core/pre-core (HBeAg) proteins, but not HBx and multiple cellular proteins tested, I would like to suggest the authors should consider to more specifically state that aerobic glycolysis modulates viral protein synthesis (or translation), instead of “viral biosynthesis”.

2. The authors stated that activation of PMK2 selectively reduces the levels of viral proteins via pentose phosphate pathway. Please provide more clear interpretation. It will be great if he authors can speculate how the viral protein synthesis (translation) is selectively modulated by aerobic glycolysis metabolism in Discussion.

3. Although the authors more carefully measured the effects of PMK2 activation on the levels of three viral proteins, the effects on viral DNA replication (intracellular core DNA), as requested by multiple reviewers, have not been measured in either pHBV3.6 transfected Huh7 cells or HBV infected HepG2-NTCP cells. Because of the unique translation mechanism of viral DNA polymerase, it will be very interesting to know if its synthesis is differentially regulated. Obviously, if the synthesis of DNA polymerase is inhibited, viral DNA synthesis should be consequentially reduced. Therefore, this assay could be an indirect measurement of aerobic glycolysis regulation of DNA pol translation.

**Part II – Major Issues: Key Experiments Required for Acceptance**

Reviewer #1: None

Reviewer #2: (No Response)

Reviewer #3: No.

**Part III – Minor Issues: Editorial and Data Presentation Modifications**

Reviewer #1: None

Reviewer #2: (No Response)

Reviewer #3: No.

PLOS authors have the option to publish the peer review history of their article (what does this mean?). If published, this will include your full peer review and any attached files.

Reviewer #1: No

Reviewer #2: No

Reviewer #3: No

Figure Files:

Data Requirements:

Reproducibility:

References:

---

## [Editor Report · Decision Letter 2]

26 Feb 2021

Dear Dr. Wang,

We are pleased to inform you that your manuscript 'Aerobic glycolysis supports hepatitis B virus protein synthesis through interaction between viral surface antigen and pyruvate kinase isoform M2' has been provisionally accepted for publication in PLOS Pathogens.

Best regards,

Aleem Siddiqui, Ph.D.

Associate Editor

PLOS Pathogens

Jing-hsiung James Ou

Section Editor

PLOS Pathogens

Kasturi Haldar

Editor-in-Chief

PLOS Pathogens

orcid.org/0000-0001-5065-158X

Michael Malim

Editor-in-Chief

PLOS Pathogens

orcid.org/0000-0002-7699-2064
---

## [Editor Report · Acceptance letter]

10 Mar 2021

Dear Dr. Wang,

We are delighted to inform you that your manuscript, "Aerobic glycolysis supports hepatitis B virus protein synthesis through interaction between viral surface antigen and pyruvate kinase isoform M2," has been formally accepted for publication in PLOS Pathogens.

Best regards,

Kasturi Haldar

Editor-in-Chief

PLOS Pathogens

orcid.org/0000-0001-5065-158X

Michael Malim

Editor-in-Chief

PLOS Pathogens

orcid.org/0000-0002-7699-2064